# RETHINKING SHAPLEY VALUE FOR NEGATIVE INTERACTIONS IN NON-CONVEX GAMES

**Wonjoon Chang, Myeongjin Lee**
Korea Advanced Institute of Science and Technology (KAIST)
{one_jj, lmjk311}@kaist.ac.kr

**Jaesik Choi**[*]
Korea Advanced Institute of Science and Technology (KAIST), INEEJI
jaesik.choi@kaist.ac.kr

## ABSTRACT

We study causal interactions for payoff allocation in cooperative game theory, including quantifying feature attribution for deep learning models. Most feature attribution methods mainly stem from the criteria of the Shapley value, which assigns fair payoffs to players based on their expected contribution in a cooperative game. However, interactions between players in the game do not explicitly appear in the original formulation of the Shapley value. In this work, we reformulate the Shapley value to clarify the role of interactions and discuss implicit assumptions from a game-theoretical perspective. Our theoretical analysis demonstrates that when negative interactions exist—common in deep learning models—the efficiency axiom can lead to the undervaluation of attributions or payoffs. We suggest a new allocation rule that decomposes contributions into interactions and aggregates positive parts for non-convex games. Furthermore, we propose an approximation algorithm to reduce the cost of interaction computation which can be applied to differentiable functions such as deep learning models. Our approach mitigates counterintuitive attribution outcomes observed in existing methods, ensuring that features critical to a model's decision receive appropriate attribution.

## 1 INTRODUCTION

As black-box models like Deep Neural Networks (DNNs) become increasingly prevalent, providing the cause of their decision-making process is crucial for model reliability and interpretation. One fundamental approach to understanding such prediction models is to quantify the attributions of individual input features to model decisions. Theoretically, feature attribution methods are usually grounded in the Shapley value (Shapley, 1953), which provides a fair payoff allocation rule in cooperative game theory. Shapley (1953) introduced a set of axioms—*linearity*, *dummy*, *symmetry*, and *efficiency*—that uniquely defines this payoff allocation. This axiomatic approach has been widely extended to machine learning research and attribution methods (Bach et al., 2015; Sundararajan et al., 2017; Rozemberczki et al., 2022).

Despite this game-theoretic foundation, there has been limited exploration of *interaction* between features and the relationship with axioms in computing attributions. Causal interaction measures the discrepancy of a player's effect on the game output depending on whether another player participates or not (VanderWeele, 2015; Keele & Stevenson, 2021). In complex games or functions, understanding the impact of interactions is crucial for assigning reasonable payoffs or attributions. However, the original formulation of the Shapley value is based on expected contributions and does not explicitly account for interactions between players (Grabisch & Roubens, 1999; Procaccia et al., 2014; Kumar et al., 2021).

In this work, we reformulate the Shapley value in terms of interaction components to clarify its dependence on interactions. Our derivation shows that the Shapley value can be interpreted as the

---
[*]Corresponding author

sum of an individual effect and its weighted interactions with other players. This reformulation demonstrates that the Shapley value implicitly assumes non-negative interactions, a property that holds when the game is convex. From a game-theoretic perspective, this assumption aligns with the efficiency axiom and the rational behavior of players within the grand coalition (Von Neumann & Morgenstern, 1944; Roth, 1988; Peters, 2015). Consequently, the Shapley value becomes a reasonable and well-defined allocation when a game is convex, and we prove that convexity is equivalent to non-negative interactions.

However, many real-world scenarios involve non-convex games, where negative interactions exist. In such cases, the Shapley value may not be a reasonable allocation derived from players' efficient and rational behaviors. Most axiomatic approaches for feature attribution, including extensions of the Shapley value, conventionally adhere to the efficiency axiom, even though many black-box models like DNNs do not satisfy the convexity. Enforcing the efficiency axiom for non-convex games leads to the undervaluation of payoffs caused by negative interactions even though the players have a potential role in improving the output. In the case study, we present illustrative examples demonstrating their impact on non-convex functions, such as a max function and a sigmoid function, and their connection to deep learning models. Negative interactions can result in the undervaluation of attributions, where highly relevant features receive low attribution scores. We discuss that these negative interactions between players arise from role redundancy or output saturation in cooperative game setups. Conventional approaches attempt to mitigate this by applying heuristic post-processing, such as taking the absolute value of attributions, but these solutions lack theoretical justification.

Building on our theoretical analysis of interaction, we extend the Shapley value to non-convex games while keeping its philosophy that assigns payoffs by measuring synergistic interactions between players. We propose a new allocation rule that decomposes each contribution into interactions and aggregates the refined interactions for non-convex games. Aggregated Positive Interactions (API), which aggregates only positive interactions, effectively resolves the undervaluation issue and ensures that crucial features receive sufficiently large attributions in non-convex games while following the Shapley value properties in convex games. Furthermore, we provide an unbiased estimation of API using a permutation sampling approach and introduce an approximation algorithm that can be applied to differentiable functions, making it computationally tractable. In practice, our approach successfully addresses the undervaluation issue in deep learning tasks.

## 2 PRELIMINARIES

### 2.1 COOPERATIVE GAME

A *cooperative game* consists of a set of players $N = \{1, \cdots, n\}$, called the *grand coalition*, and a *characteristic function* $v : 2^N \to \mathbb{R}$, which maps a coalition $S \subseteq N$ to the utility $v(S)$ players in $S$ achieve. The goal is to determine the payoff vector $\phi(v) \in \mathbb{R}^n$ where $i$-th element $\phi_i(v)$ indicates the payoff allocated to player $i$ from the total utility $v(N)$. This game is sometimes called a *transferable utility* game allowing the utility to be fully transferred to the players as their payoffs (Von Neumann & Morgenstern, 1944; Roth, 1988; Peters, 2015). In classic literature, for convenience, the function $v$ itself is often referred to as the game, and the cardinality of each set of players $(N, S, T, \cdots)$ is denoted by the corresponding lower-case letter $(n, s, t, \cdots)$. We follow this convention in this work.

A game is *convex* if

$$v(S) + v(T) \leq v(S \cup T) + v(S \cap T), \quad \forall S, T \subseteq N. \tag{1}$$

If Equation (1) holds under the more relaxed condition $S \cap T = \emptyset$, then the game is *super-additive*. This indicates that a convex game is a special case of a super-additive game.

**Remark.** *The implicit assumption in a cooperative game is that players strategically form the grand coalition to maximize their payoffs (Roth, 1988; Fujimoto et al., 2006; Peters, 2015). Convex games indicate that the benefits of joining a coalition increase as the coalition size grows, ensuring that cooperation always leads to higher utility and that forming the grand coalition $N$ is an optimal strategy (Shapley, 1971).*

## 2.2 SHAPLEY VALUE

The Shapley value (Shapley, 1953) is the payoff vector $\phi(v) \in \mathbb{R}^n$ that fairly distributes the total utility $v(N)$ to each player $i \in N$:

$$\phi_i(v) = \sum_{S \subseteq N \setminus \{i\}} \frac{1}{n} \binom{n-1}{s}^{-1} [v(S \cup \{i\}) - v(S)]. \tag{2}$$

It is a unique form that satisfies four axioms designed for a fair allocation: *linearity*, *dummy*, *symmetry*, and *efficiency* (Weber, 1988). The detailed explanations of the axioms are provided in Appendix A. Equation (2) can be interpreted as the weighted average of $i$'s contribution when joining all possible coalitions. Furthermore, it can be viewed as the expectation of the causal effects on the function output since it evaluates the expected output increase while putting or removing a player (Janzing et al., 2020; Pearl, 2022). From this perspective, $\Delta_i v(S) := v(S \cup \{i\}) - v(S)$ is referred to as *(causal) effect* or *(marginal) contribution*.

An alternative formulation of the Shapley value is represented as the expected contribution over all possible random orderings. Let $\Pi(N)$ denote the set of all permutations of the player set $N$. For a given $\pi \in \Pi(N)$, define $\pi^i$ as the set of players preceding player $i$ in the ordering $\pi$. Then, the Shapley value can be expressed as follows:

$$\phi_i(v) = \frac{1}{n!} \sum_{\pi \in \Pi(N)} \left( v(\pi^i \cup \{i\}) - v(\pi^i) \right). \tag{3}$$

## 2.3 EFFICIENCY IN COOPERATIVE GAMES

The Shapley value is a well-defined solution that belongs to the *core* (Shapley, 1971; Dehez, 2017). The *core* of a game is a set of payoff vectors $a \in \mathbb{R}^n$ that satisfy the following two conditions:

$$\{a \in \mathbb{R}^n \mid \sum_{i \in N} a_i = v(N), \sum_{i \in S} a_i \geq v(S) \; \forall S \subset N\}. \tag{4}$$

The core represents the socially stable outcomes where no partial coalition can achieve a better result than the grand coalition. The first equality condition states that the sum of individual payoffs should equal the total utility $v(N)$. This equality condition is the same as the *efficiency* axiom of the Shapley value. The second inequality condition, namely the *coalitional rationality*, implies that the sum of payoffs allocated to any subset coalition $S$ is at least as large as the utility $v(S)$ the coalition $S$ can achieve on its own.

The concept of efficiency originates from *Pareto efficiency*, which states that no player's payoff can be improved without reducing another player's payoff (Peleg & Sudhölter, 2007; Peters, 2015). This implicitly assumes that all players prefer to maintain cooperation through their rational behaviors. In this background, it is natural to consider rationality, which is linked to the core, when imposing efficiency for payoff allocation in cooperative games. Notably, the Shapley value is a solution positioned at the center of the core when the core exists (Shapley, 1971).

## 3 INTERACTIONS IN SHAPLEY VALUE

The Shapley value assesses a player's payoff based on the average causal effect across all possible coalitions. However, when effects vary significantly depending on the context, the resulting payoff allocation may be distorted. This study highlights the impact of interactions inherently captured in computing the Shapley value and argues that its underlying assumptions may not be suitable for games with complex interactions.

### 3.1 REFORMULATION OF SHAPLEY VALUE

We first define *interaction* between two players in the cooperative game.

**Definition 1** (interaction). For a cooperative game $v$ on a player set $N$, an interaction between two player $i, j \in N$ given a subset of players $T \subseteq N \setminus \{i, j\}$ is defined as follows:

$$\begin{aligned} I_{ij}(T) &= v(T \cup \{i, j\}) - v(T \cup \{i\}) - v(T \cup \{j\}) + v(T) \\ &= \Delta_i v(T \cup \{j\}) - \Delta_i v(T). \end{aligned} \tag{5}$$

This definition comes from the notion of causal interaction in the causal inference literature (VanderWeele, 2015; Keele & Stevenson, 2021). It quantifies the discrepancy of player $i$'s effect on the output depending on whether player $j$ participates or not. In the absence of interactions among players, each player's effect remains constant regardless of others' participation, making payoff allocation trivial. However, cooperative games typically assume synergistic interactions, where players cooperate to maximize their payoffs.

Despite this, the original formulation of the Shapley value provides limited insight into how such interactions influence payoff allocation. To address this limitation, we reformulate the Shapley value to explicitly incorporate interaction terms, leading to the following representation.

**Theorem 1.** *The Shapley value is a weighted sum of interactions:*

$$\phi_i(v) = \Delta_i v(\emptyset) + \sum_{t=0}^{n-2} \frac{1}{n} \binom{n-1}{t}^{-1} \sum_{\substack{j \in N \\ j \neq i}} \sum_{\substack{T \subseteq N \setminus \{i,j\} \\ |T|=t}} I_{ij}(T). \tag{6}$$

*Proof.* See Appendix. □

The theorem states that the Shapley value measures the sum of the individual effect and the average interactions resulting from cooperation.

## 3.2 RELATIONSHIP WITH CONVEXITY

In Section 2.1, we introduce the concept of a convex game, where the benefits of cooperation increase as the coalition size grows. The connection between convexity and interactions can be revealed from the following theorem, which states a game is convex if and only if all interactions are non-negative.

**Theorem 2.** *A game $v$ is convex if and only if $I_{ij}(R) \geq 0 \; \forall i, j \in N, \forall R \subseteq N \setminus \{i,j\}$.*

*Proof.* See Appendix. □

Convexity is closely related to the core described in Section 2.3. It guarantees the existence of the core and ensures that it forms a polytope in the solution space, with its vertices corresponding to the *marginal contribution vectors* (Shapley, 1971; Ichiishi, 1981; Dehez, 2017). For a given permutation $\pi \in \Pi(N)$, the *marginal contribution vector* is an $n$-dimensional vector, where the $i$-th element is given by $v(\pi^i \cup \{i\}) - v(\pi^i)$ with $\pi^i$ representing the set of players preceding player $i$ in the ordering $\pi$. The alternative formulation in Equation (3) shows that the Shapley value is obtained as the average of these marginal contribution vectors. Theorem 1 follows from decomposing marginal contributions into the sum of cumulative interactions (Appendix C.1). Therefore, in convex games, positive interactions arising from player cooperation enhance marginal contributions, leading to increasing the Shapley value.

## 3.3 PROBLEMS IN NON-CONVEX GAMES

Non-convex games imply the existence of negative interactions between players. In such cases, the core does not encompass all marginal contribution vectors, indicating that some marginal contributions arise from non-rational behaviors. As a result, players' payoffs can be underestimated, even when they have the potential to enhance the overall outcome. This raises the question of whether the Shapley value remains an appropriate allocation rule in non-convex settings. The *efficiency* (or *completeness*) axiom is conventionally applied in the applications of the Shapley value and attribution methods for black-box models (Lundberg, 2017; Sundararajan et al., 2017). This axiom ensures that the solution, including the Shapley value, is expressed as an expectation of marginal contribution vectors (Weber, 1988). Enforcing the efficiency axiom for non-convex games can lead to the undervaluation of payoffs or feature attributions caused by negative interactions even though the players have a potential role in improving the output. We can identify the issue of undervaluation that negative interactions explicitly diminish a player's payoff in Equation (6).

Negative interactions between players can arise from various factors, such as personal or emotional conflicts. However, in cooperative games, where the primary assumption is that players collaborate to maximize total utility and individual payoffs, it is more reasonable to attribute such negative interactions to functional role redundancy. This concept extends to feature attribution in machine learning. Black-box models typically leverage all available features—forming a grand coalition—to make decisions without regard for their conflicts. However, high-capacity models like DNNs usually exhibit non-convexity, leading to negative interactions among features. This has been linked to unexpected experimental results in Shapley-based feature attribution methods, where features crucial for decision-making may receive low attribution scores. To mitigate this, conventional approaches often apply heuristic post-processing adjustments, such as taking the absolute value of attributions.

To better understand the implications of negative interactions, Section 4 presents illustrative examples demonstrating their impact on non-convex functions, including deep learning models. Section 5 then introduces a principled game-theoretic approach to address these challenges.

## 4 CASE STUDY

In this section, we will explore examples of non-convex games where negative interactions occur. The problem emerges when the grand coalition is no longer driven by rational behavior. If the grand coalition is inefficient—meaning that some players can contribute significantly to the outcome but are not essential for maximizing the final output—they may experience negative interactions. In cooperative games, negative interactions between players can happen due to their role redundancy or output saturation rather than their personal conflicts, which we illustrate through examples such as a max function and a sigmoid function. Furthermore, we discuss how negative interactions naturally arise in deep learning models that incorporate these functions.

**Max function.** The max function returns the highest value that the participating players can yield. A player's causal effect on the max function output diminishes when another player with a similar contribution joins. In this case, interaction becomes negative, and indeed the max function is non-convex. Furthermore, consider the following function

$$f(x_1, \cdots, x_8) = \max(x_1, x_2, x_3, 4x_4) + \max(6x_5, 6x_6, 6x_7, 7x_8).$$

We set each variable $x_i$ to be binary (0 or 1) to represent the participation keeping the setup of cooperative game theory. We can generally expect that player 8 may get a larger payoff than player 4. However, there are negative interactions in this function, for example, $I_{14}(\emptyset) = -1, I_{58}(\emptyset) = -6$, which lead to counterintuitive results. The Shapley value of player 8 is $10/4$ while that of player 4 is $13/4$. This phenomenon conceptually arises from redundancy in the roles of individual players. Even if some player has a large impact on the outcome, their interactions become negative when their roles overlap, leading to underrated payoffs. In Appendix D, this issue is intuitively described in an image recognition scenario.

**Sigmoid function.** Another example of a non-convex function that yields problematic negative interactions is the sigmoid function that takes the sum of features as input. There exist negative interactions when the input becomes sufficiently large, pushing the function into its near-saturation region. Consider the following function

$$f(x_1, x_2, x_3, x_4) = 10 \cdot \sigma(7x_1 + 6x_2) + 10 \cdot \sigma(2x_3 + 0.1x_4)$$

where $\sigma$ denotes the sigmoid function $1/(1 + e^{-x})$. Similar to the max function case, players 1 and 2 exhibit a stronger negative interaction ($I_{12}(\emptyset) = -4.96$) compared to players 3 and 4 ($I_{34}(\emptyset) = -0.15$). Consequently, player 1's Shapley value (2.51) is lower than player 3's (3.73). This phenomenon arises due to the presence of saturation. More generally, this issue extends to any squashing function with saturation or an upper-bound, such as the softmax function.

**Deep Learning model.** A deep learning model is a non-convex function, incorporating various non-convex components within its architecture. Consider *max pooling* in Convolutional Neural Networks (CNNs) and *attention mechanisms* in Transformer models. CNN classifiers commonly utilize the max pooling as a fundamental operation and attention mechanisms in Transformers compute a weighted sum of values, where the weights are derived from the softmax scores of query-key similarities (Vaswani et al., 2017). Given these structural properties, similar issues arising from

non-convexity can be expected in CNNs and Transformers. In Figures 1 and 3, we show the counter-intuitive results of the Shapley value in such models, highlighting the impact of negative interactions on feature attributions.

## 5  AGGREGATED POSITIVE INTERACTIONS

We have discussed the issue of utilizing the Shapley value and its variants for non-convex games. To address this, we propose a generalized solution that decomposes each contribution into interactions and aggregates them using a refinement function $g : \mathbb{R} \to \mathbb{R}$:

$$\phi_i(v) = \Delta_i v(\emptyset) + \sum_{t=0}^{n-2} \frac{1}{n} \binom{n-1}{t}^{-1} \sum_{\substack{j \in N \\ j \neq i}} \sum_{\substack{T \subseteq N \setminus \{i,j\} \\ |T|=t}} g(I_{ij}(T)). \tag{7}$$

The function $g$ represents how interactions are interpreted as attributions to the output. When $g$ is an identity function, interactions are directly used as attributions in the calculation, regardless of their magnitude or sign, resulting in the Shapley value.

To alleviate the undervaluation in non-convex games, we suggest a solution with $g(x) = \max(x, 0)$, namely *Aggregated Positive Interactions (API)*. It can be interpreted as the extension of the Shapley value to non-convex games. In convex games, API equals the Shapley value, as all interactions are preserved. In non-convex games, API evaluates a player's potential contribution to improving the output by forming synergistic coalitions by filtering out problematic negative interactions. Since negative interactions are removed in aggregation, a player's payoff is no longer underrated from irrational coalitions. API effectively addresses the undervaluation demonstrated in Section 4. For the max function case, API assigns values of 7 and 4 to players 8 and 4, respectively, and for the sigmoid function case, it assigns values of 4.99 and 3.81 to players 1 and 3, respectively.

**Axioms.** In non-convex games, a solution does not need to strictly adhere to the *efficiency* axiom due to the issues described in Sections 3.3 and 4. Enforcing efficiency ensures that the total sum of attributions remains fixed, which inevitably leads to lower attributions for some players. In this spirit, API relaxes the efficiency axiom in non-convex games to prevent undervaluation caused by negative interactions. The *dummy* axiom states that if a player has no causal effect on the function output then the corresponding attribution should be zero. Any solution based on aggregated interactions with $g(0) = 0$ satisfies the dummy axiom. We further discuss the converse of this axiom: does zero attribution imply that the corresponding player has no meaningful contribution to improving the output? In the Shapley value and its variants, which follow the form of expected contribution, players in non-convex games may receive zero attribution due to negative interactions canceling out their positive contributions. This can result in undervaluation, even when a player has a potential role in improving the output. In contrast, API ensures that zero attribution genuinely reflects a player's inability to enhance the output (strictly speaking, in 0-normalized games). Consequently, players with higher attributions in API are those who significantly contribute to improving the output.

The choice of $g$ depends on how the user interprets interactions as attributions in a given task. In Appendix E.2, we additionally provide attribution results using different choices of $g$, specifically Leaky ReLU and Softplus. The solutions derived from these functions do not follow all the properties that API upholds. However, in scenarios where a decrease in the function output carries meaningful implications, such as in regression problems, strictly eliminating all negative interactions could misrepresent the influence of certain variables, and they should be carefully handled. Therefore, the treatment of negative interactions should be determined based on the context of the application. Especially, API focuses on solving the undervaluation problems derived from role redundancy or output saturation described in Section 4 by removing unexpected negative interactions and preserving positive ones.

### 5.1  ESTIMATION THROUGH PERMUTATION SAMPLING

Since dealing with interaction terms requires high computational cost for applying complex black-box models like DNNs, we first propose the estimation of Equations (6) and (7). Then, we introduce an approximation algorithm to reduce the computational cost for interaction terms using gradient information.

We reformulate Equation (6) into the expectation form to enable sample estimation with vectorized representations. Let $i \sim u_1(S)$ be the uniform distribution of player $i$ from set $S$, and $T \sim u_2(t, S)$ be the uniform distribution of subset $S$ with cardinality $|T| = t$. $\phi(v)$ and $\Delta v(\emptyset)$ denote the Shapley value and the marginal contributions of all players at the empty set in a vector form, respectively. We define an interaction vector $I_{\cdot j}(T) \in \mathbb{R}^n$ where the $i$-th element is $I_{ij}(T)$ for given $T \subseteq N \setminus \{j\}$. We set $I_{ij}(T) = 0$ when $i = j$ or $i \in T$, then $I_{\cdot j}(T)$ is well-defined for any $T \subseteq N \setminus \{j\}$. Then, we obtain the following vectorized form to estimate the Shapley value with interaction vectors.

**Theorem 3.** *The payoff vector of the Shapley value $\phi(v)$ is represented as follows:*

$$\phi(v) = \Delta v(\emptyset) + \sum_{t=0}^{n-2} \mathbb{E}_{j \sim u_1(N), T \sim u_2(t, N \setminus \{j\})}[I_{\cdot j}(T)]. \tag{8}$$

*Proof.* See Appendix. $\qquad\square$

We can now apply an unbiased sample estimation for the expected interaction at each cardinality and sum over interactions. This estimation can be also conducted by permutation sampling. For each permutation, we sequentially add features, and compute interactions at each level of cardinality. In this setup, we need to reformulate Equation (8) with the expectation under the uniform distribution of permutations.

**Corollary 1.** *The payoff vector of the Shapley value $\phi(v)$ is represented as follows:*

$$\phi(v) = \Delta v(\emptyset) + \sum_{t=0}^{n-2} \mathbb{E}_{\pi \sim \mathrm{Unif}(\Pi(N))}[I_{\cdot \pi_{t+1}}([\pi]_t)]. \tag{9}$$

*where $\Pi(N)$ is a collection of all possible permutations of players in $N$. $\pi_t$ represents the $t$-th element in $\pi$, and $[\pi]_t$ represents the subset of players up to the $t$-th player in the ordering $\pi$.*

It can be easily proved by Theorem 3. From the result, we can apply permutation sampling to estimate the Shapley value with interactions. It can be interpreted as the extension of permutation sampling on the original Shapley value (Castro et al., 2009). Simply, we obtain Aggregated Positive Interactions by maintaining only non-negative interactions during sampling.

## 5.2 APPROXIMATION ALGORITHM

The major computational burden comes from the interaction terms. Since the computational cost for interactions grows with the number of player combinations, it requires $n$-times more computations compared to computing individual effects in the original Shapley value. We propose an algorithm to reduce the cost of interaction computation that can be applied to differentiable functions. In differentiable functions like DNNs, this approach leverages backpropagation to reduce computational complexity effectively.

**Notations.** We first explain notations for our algorithm described in Algorithm 1. $v$ is a differentiable function to assess attributions, such as a deep learning predictor. $x$ is a target instance with $n$ features, and each feature is treated as a player in $N$. A baseline value in $\bar{x}$ refers to the value assigned when a feature does not participate in the game, and it is typically set to the mean value from the data or zero. $\nabla v(S)$ denotes a gradient of $v$ evaluated when features in $S$ are set to values in $x$ and the others are set to values in $\bar{x}$.

The original definition of interaction is expressed as the difference between two contributions, represented by discrete derivatives. We replace this discrete derivative with a partial derivative using a first-order Taylor approximation. This allows us to compute interactions quickly by leveraging gradients obtained through backpropagation.

$$I_{ij}(T) = \Delta_i v(T \cup \{j\}) - \Delta_i v(T) \approx \{\partial_i v(T \cup \{j\}) - \partial_i v(T)\} * (x_i - \bar{x}_i) \tag{10}$$

When calculating interactions for all features given a set $T$, the original computation requires $O(n^2)$ forward passes. However, with this approximation, all feature gradients can be computed simultaneously with a single backpropagation, reducing the computation to $O(n)$ backward passes. By utilizing this approximation and permutation sampling, our algorithm computes API as described in Algorithm 1. The practical runtime of our algorithms is further discussed in Appendix E.3.

---

**Algorithm 1** Approximation for Aggregated Positive Interactions

---

    **Input:** Differentiable function $v$, Instance $x$, Baseline $\bar{x}$
    **Parameter:** # of permutations $k$
    **Output:** Attribution $\phi$
    Initialize $\text{Cnt} \leftarrow 0$, $\phi \leftarrow \Delta v(\emptyset)$, $g_1 \leftarrow \nabla v(\emptyset)$
    **while** $\text{Cnt} < k$ **do**
        sample $\pi \sim \text{Unif}(\Pi(N))$
        **for** $t = 0, \cdots, n-2$ **do**
            $g_2 \leftarrow \nabla v([\pi]_{t+1})$
            $\hat{I} \leftarrow \max((g_2 - g_1) \circ (x - \bar{x}), \mathbf{0}_n)$
            $\hat{I}_i \leftarrow 0 \quad \forall i \in [\pi]_{t+1}$
            $\phi \leftarrow \phi + \hat{I}/k$
            $g_1 \leftarrow g_2$
            $\text{Cnt} \leftarrow \text{Cnt} + 1$
        **end for**
    **end while**
    **return** $\phi$

---

# 6 EVALUATION

In Section 4, we introduced some simple non-convex examples to identify the impact of negative interactions on the Shapley value. In this section, we identify such phenomenon in practical deep learning tasks and evaluate the results from Aggregated Positive Interactions (API). Our experiments are conducted on image classifiers (VGG19 (Simonyan & Zisserman, 2014), ResNet50 (He et al., 2016)) trained on the ImageNet dataset (Deng et al., 2009), and a sentence classifier (BERT (Devlin, 2018)) trained on the IMDB Review dataset (Maas et al., 2011). In experimental results, Approximated Shapley value (abbreviated as Approx. SV) denotes the approximation of the Shapley value by sampling permutation and aggregating all interactions. We samples 100 permutations for the image classifiers and 300 permutations for the sentence classifier. In the case of ImageNet data, we convert images into $20 \times 20$ patches for feasible computation.

## 6.1 IMPACT OF POSITIVE INTERACTIONS ON IMAGE CLASSIFIER

We analyze the impact of positive interactions on VGG19. Figure 1 shows attribution results, presenting high-attribution regions, and low-attribution regions when aggregating all interactions versus only positive interactions. In the attribution map, red indicates positive attribution, white represents zero attribution, and blue denotes negative attribution.

When all interactions are aggregated, attributions tend to become more dispersed. This dispersion can be explained by negative interactions, as seen in max pooling, where nearby inputs are grouped together, creating negative interactions within the same pooling operator. The Shapley value struggles to pinpoint key contributing areas to the model decision. However, when we aggregate only positive interactions, key regions become more distinct, such as the eyes, nose, and front paws of a red panda or the wheels and windows of a sports car.

## 6.2 UNDERRATED ATTRIBUTIONS FROM EFFICIENCY AXIOM

To demonstrate the limitations of methods based on the efficiency axiom in non-convex games, we compare our approach to relevant attribution methods using ResNet50 in Figure 2. For KernelSHAP (Lundberg, 2017), we use 40,000 samples, as our method approximates using 100 permutations across 400 features. Integrated Gradients (IG) (Sundararajan et al., 2017) is computed with 100 steps, and the attributions are summed for all pixels within each patch. We evaluate performance using the Insertion and Deletion metrics. Insertion measures the logit value when the top 30% attribution patches are added, while Deletion measures it when the bottom 30% attribution patches are removed. Higher values for both metrics indicate better identification of regions crucial for the model's decision.

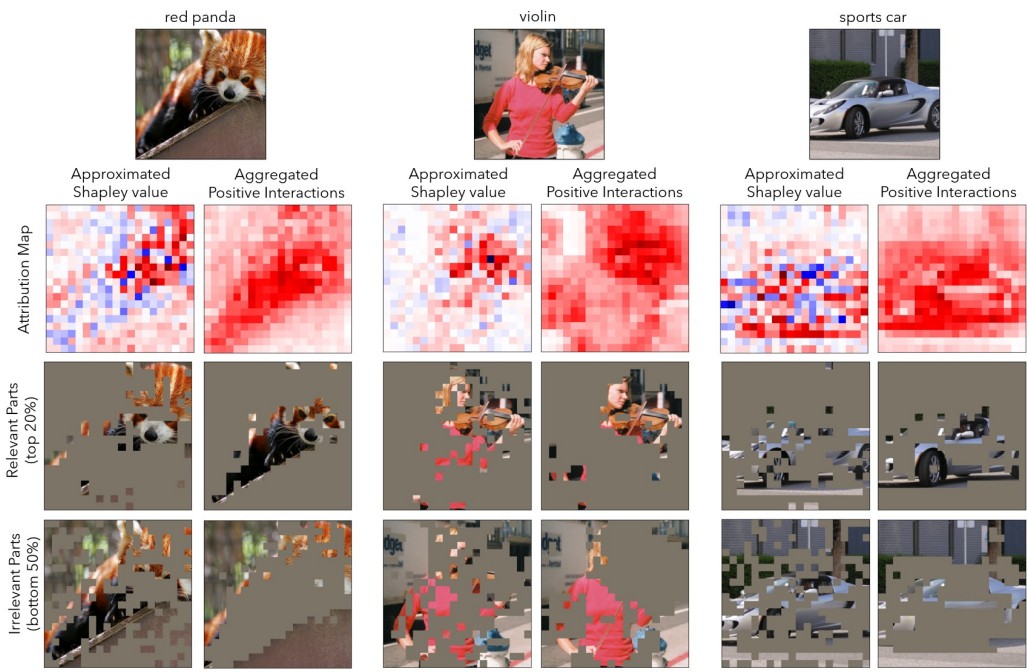

Figure 1: Aggregated Positive Interactions on the ImageNet dataset. While the approximated Shapley Value produces dispersed attributions across relevant regions, API more effectively captures these regions.

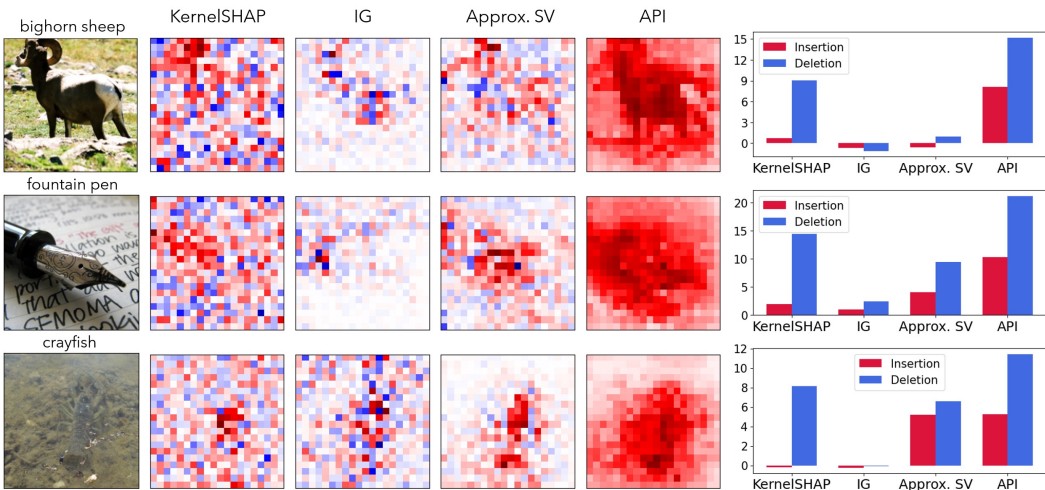

Figure 2: Comparison to other Attribution methods based on Efficiency axiom. Insertion measures the logit value when the top 30% attribution patches are added, while Deletion measures it with the bottom 30% removed. API achieves better results compared to the previous methods.

KernelSHAP and IG, constrained by the efficiency axiom, tend to disperse attributions across various regions, similar to the Approximated Shapley value. In contrast, API focuses attributions on key input features, avoiding unnecessary assignments to the background. In practice, absolute attribution values are often used heuristically for better interpretability. API addresses this issue with a more principled, game-theoretic approach. While KernelSHAP sometimes performs better numerically, it still suffers from attribution dispersion in irrelevant regions. This may stem from errors introduced by the first-order Taylor approximation used in API for interaction computation.

For additional quantitative results, we assess effective complexity (Nguyen & Martínez, 2020) in Appendix E.1. A low effective complexity indicates that the attribution provides a more compact explanation by focusing on a smaller number of significant features. On average, API achieves 30% lower complexity than the other methods.

## 6.3    APPLICATION ON LANGUAGE MODEL

We demonstrate the application of API on a language model, using a BERT (Devlin, 2018) trained on the IMDB dataset for sentence prediction (Maas et al., 2011). This model takes a movie review, which is a sequence of tokens, and predicts whether it is positive or negative. Figure 3 shows a fraction of input texts where the top row is predicted as negative, while the bottom row is positive. For these texts, we compare the approximated Shapley value and API, estimated with 300 permutations, by highlighting the tokens with attribution in the top 20%.

In the negative review case, the approximated Shapley value only highlights 'hardly be' while API focuses on the entire phrase 'can hardly be taken seriously on any level'. This indicates that the portion from 'taken' to 'level' was underrated due to strong negative interactions. While 'hardly be' alone does not intuitively seem influential for the negative prediction, it conveys a negative meaning when combined with the subsequent tokens from 'taken' to 'level'. These tokens have strong negative interactions that obscure their contributions, which API correctly addresses by focusing on the key phrase. A similar issue arises in the positive review case in the bottom row, where the Shapley value underrates tokens 'i', 'liked', and 'movie', while assigning higher attributions to other irrelevant tokens. API effectively resolves this by properly attributing the entire phrase 'i really liked this movie'.

|  | Approximated Shapley value | Aggregated Positive Interactions |
|---|---|---|
| Negative Review | it doesn' t matter what one' s political views are because this film can hardly be taken seriously on any level. | it doesn' t matter what one' s political views are because this film can hardly be taken seriously on any level. |
| Positive Review | i really liked this movie. i' ve read a few of the other comments, and although i pity those who did not understand it, i do agree with some of the criticisms. | i really liked this movie. i' ve read a few of the other comments, and although i pity those who did not understand it, i do agree with some of the criticisms. |

Figure 3: Application on a Language Model. While the approximated Shapley value assigns low attributions to the tokens that seem relevant due to the negative interactions, API effectively results in more reasonable attributions by focusing only on the positive interactions.

## 7    CONCLUSION

This study has explored the impact of causal interactions on payoff allocation in cooperative game theory, with a particular focus on feature attribution in deep learning models. We reformulate the Shapley value to account for interactions between players, and demonstrate that these interactions explicitly influence payoffs in the Shapley value framework. From the game-theoretic perspective, we highlight that this framework implicitly assumes players' rational and efficient behaviors to improve the total utility, which may conflict with the presence of negative interactions in non-convex games. In such games, the efficiency axiom of the Shapley value can unintentionally lead to the undervaluation of payoffs or attributions. To extend the Shapley value to non-convex games while preserving its underlying philosophy, we propose a new allocation rule, namely Aggregated Positive Interactions (API), which decomposes contributions into interactions and aggregates positive ones. Additionally, we introduce an approximation algorithm to enhance computational tractability in interaction computation for differentiable functions. Our experimental results show that API effectively resolves the counterintuitive results of the Shapley value and feature attribution methods based on the efficiency axiom. We expect that our work will serve as a solid foundation for extending game-theoretic approaches to the attribution task, contributing to the interpretability of deep learning models.

ACKNOWLEDGMENTS

This work was supported by Hyundai Motor Company, which provided valuable guidance on the research direction, and by Institute for Information & communications Technology Planning & Evaluation (IITP) grant funded by the Korea government (MSIT) (No. RS-2019-II190075, Artificial Intelligence Graduate School Program (KAIST); No. RS-2022-II220984, Development of Artificial Intelligence Technology for Personalized Plug-and-Play Explanation and Verification of Explanation; No. RS-2024-00457882, AI Research Hub Project).

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

# APPENDIX

## A  AXIOMS FOR SHAPLEY VALUE

The Shapley value (Shapley, 1953) is the payoff vector $\phi(v) \in \mathbb{R}^n$ that fairly distributes the total utility $v(N)$ to each player $i \in N$:

$$\phi_i(v) = \sum_{S \subseteq N \setminus \{i\}} \frac{1}{n} \binom{n-1}{s}^{-1} [v(S \cup \{i\}) - v(S)]. \tag{11}$$

It is a unique form that satisfies four axioms designed for a fair allocation: *efficiency*, *symmetry*, *linearity*, and *dummy* (Weber, 1988; Peleg & Sudhölter, 2007; Dehez, 2017; Rozemberczki et al., 2022).

**Efficiency.**  The sum of all payoffs equals the total utility of the grand coalition:

$$\sum_{i \in N} \phi_i(v) = v(N). \tag{12}$$

It is often called *Pareto efficiency* (or *Pareto optimality*), which assumes that players behave efficiently: no partition of players can achieve a larger total utility than the grand coalition. Under this assumption, the grand coalition results from the players' optimal strategies. Therefore, it is reasonable to directly allocate the utility of the grand coalition to the players' payoffs.

**Symmetry.**  Two players $i$ and $j$ are *symmetric* if $v(S \cup \{i\}) = v(S \cup \{j\})$ for every $S \subseteq N \setminus \{i, j\}$. If two players $i$ and $j$ are symmetric, their payoffs should be equal:

$$\phi_i(v) = \phi_j(v) \quad \text{if } i, j \text{ are symmetric.} \tag{13}$$

The symmetry property ensures that players with identical contributions to the utility receive the same payoff.

**Linearity.**  For any two games $(N, v), (N, w)$ and a real number $\alpha$, the payoff vector holds that

$$\phi(v + w) = \phi(v) + \phi(w), \tag{14}$$
$$\phi(\alpha v) = \alpha \phi(v). \tag{15}$$

The first property is often called *additivity*. It seems reasonable that for the rescaled game $\alpha v$ the allocation rule simply rescales a player's original payoff with the same amount.

**Dummy.**  A player $i$ is a *dummy* (or *null*) in $v$ if $v(S \cup \{i\}) = v(S)$ for every $S \subseteq N \setminus \{i\}$. If $i$ is a dummy player, the payoff of $i$ should be zero:

$$\phi_i(v) = 0 \text{ if } i \text{ is a dummy player.}$$

It is natural to assign a zero payoff to a player who does not affect the total utility in any situation.

## B  RELATED WORK

**Shapley Value.**  The payoff allocation for the utility function has been studied with axiomatic approaches in cooperative game theory literature (Roth, 1988; Fujimoto et al., 2006; Peters, 2015). The Shapley value (Shapley, 1953) is the payoff allocation that uniquely satisfies the *linearity*, *dummy*, *symmetry*, and *efficiency* axioms (Weber, 1988). It can also be interpreted as the expectation of the causal effects since the contribution of a player is measured by *do*-operator (Janzing et al., 2020; Pearl, 2022). Due to the well-defined axioms of the Shapley value, there have been a lot of approaches to apply the Shapley value to deep learning literature, especially for measuring feature attribution (Scott et al., 2017; Sundararajan & Najmi, 2020; Rozemberczki et al., 2022). Despite the widespread use of the Shapley value, some studies have identified problems with the axioms it relies on. Kumar et al. highlighted the limitation of the Shapley value when the game is not inessential and proposed Shapley Residuals to quantify the lost information when using the Shapley value. Kwon & Zou raised the inappropriateness of the efficiency axiom, pointing out the information gap depending on the size of the feature sets.

**Game-theoretic Interaction.** The concept of interaction was firstly proposed to deal with the information of *cooperation* existing among players (Grabisch & Roubens, 1999), which cannot be taken into account by the Shapley value (Shapley, 1953) and its variants (Banzhaf III, 1964; Monderer & Samet, 2002). Since the natural extensions of the axioms of the Shapley value do not guarantee the uniqueness of interaction indices, previous studies have introduced additional axioms (Grabisch & Roubens, 1999; Sundararajan et al., 2020; Tsai et al., 2023). Fumagalli et al. proposed a general form of these interaction indices and an efficient sampling-based estimator SHAP-IQ. Some approaches have attempted to explain the underlying reasoning of the inference of DNNs through the game-theoretic interactions (Deng et al., 2021; Ren et al., 2023; Li & Zhang, 2023).

**Attribution Methods.** Attribution methods aim to measure the contributions of input features to the model output. While LRP-based methods sequentially redistribute the model output across the layers in reverse order (Bach et al., 2015; Montavon et al., 2017; Shrikumar et al., 2017; Nam et al., 2020), IG and its variants evaluate attributions by aggregating gradients along one or more paths in the input space (Sundararajan et al., 2017; Kapishnikov et al., 2021; Jeon et al., 2023). Additive feature attribution methods unify several existing methods, with SHAP uniquely satisfying Shapley-based axioms (Lundberg, 2017). These methods typically impose the efficiency axiom, requiring attributions to sum to the model output.

## C  PROOF

For convenience, we represent the cardinality of $N, S, T$ as $n, s, t$, respectively. $\Pi(S)$ denotes the set of all permutations of the player set $S$. Given $\pi \in \Pi(S)$ and $i \in S$, $\pi^i$ is the set of players preceding player $i$ in the ordering $\pi$. $[\pi]_k$ denotes the subset of players up to the $k$-th player in the ordering $\pi$, where $[\pi]_0 := \emptyset$ and $[\pi]_s := S$.

### C.1  PROOF OF THEOREM 1

**Lemma 1.** *Given a player $i$ and $S \subseteq N \setminus \{i\}$, for any permutation $\pi \in \Pi(S)$ and $k \in \{1, \cdots s\}$,*

$$\Delta_i v(S) = \Delta_i v([\pi]_{k-1}) + \sum_{t=k}^{s} I_{i,\pi_t}([\pi]_{t-1}), \tag{16}$$

*where $\pi = (\pi_1, \cdots, \pi_s)$, $[\pi]_k$ represents the subset of players up to the $k$-th player in the ordering $\pi$, $[\pi]_0 := \emptyset$ and $[\pi]_s := S$.*

*Proof.* It can be simply proved by the definition of interactions.

$$\begin{aligned}
\Delta_i v(S) &= \Delta_i v([\pi]_s) \\
&= \Delta_i v([\pi]_{s-1}) + \{\Delta_i v([\pi]_s) - \Delta_i v([\pi]_{s-1})\} \\
&= \Delta_i v([\pi]_{s-1}) + I_{i,\pi_s}([\pi]_{s-1}) \\
&= \cdots \\
&= \Delta_i v([\pi]_{k-1}) + \sum_{t=k}^{s} I_{i,\pi_t}([\pi]_{t-1}).
\end{aligned} \tag{17}$$

$\square$

**Theorem 1.** *The Shapley value is a weighted sum of interactions:*

$$\phi_i(v) = \Delta_i v(\emptyset) + \sum_{t=0}^{n-2} \frac{1}{n} \binom{n-1}{t}^{-1} \sum_{\substack{j \in N \\ j \neq i}} \sum_{\substack{T \subseteq N \setminus \{i,j\} \\ |T|=t}} I_{ij}(T). \tag{18}$$

*Sketch of proof.*

1. The Shaley value is the expectation of marginal contributions over all possible random orderings.

2. Each contribution is decomposed into cumulative interactions.

3. The coefficients of individual interactions in the Shapley value equation are obtained by aggregating the results of the decomposition.

*Proof.* The Shapley value can be expressed as the expected marginal contribution over all possible random orderings:

$$\phi_i(v) = \frac{1}{n!} \sum_{\pi \in \Pi(N)} \Delta_i v(\pi^i) = \frac{1}{n!} \sum_{\pi \in \Pi(N)} \left( v(\pi^i \cup \{i\}) - v(\pi^i) \right). \tag{19}$$

Each contribution $\Delta_i v(\pi^i)$ can be decomposed into various combinations of interactions depending on the order of players in $\pi^i$. Let $\pi := (\pi_1, \cdots, \pi_s)$ be a fixed ordering for set $S$. By Lemma 1, we obtain

$$\Delta_i v(S) = \Delta_i v([\pi]_s) = \sum_{t=1}^{s} I_{i,\pi_t}([\pi]_{t-1}) + \Delta_i v(\emptyset). \tag{20}$$

By substitution, the Shapley value can be represented as the sum of an individual contribution term and the aggregated interactions between players, weighted over all possible subsets.

$$\phi_i(v) = \Delta_i v(\emptyset) + \sum_{\substack{j \in N \\ j \neq i}} \sum_{T \subseteq N \setminus \{i,j\}} w_{ij}^T I_{ij}(T) \tag{21}$$

The coefficient $w_{ij}^T$ is determined by counting the occurrences of $w_{ij}^T$ in $\Delta_i v(\pi^i)$ in Equation (19).

$$\begin{aligned} w_{ij}^T &= \sum_{\bar{\pi} \in \Pi(T)} \frac{1}{n!} \sum_{\pi \in \Pi(N)} \mathbb{1}[[\pi]_t = \bar{\pi}, \pi_{t+1} = j] \\ &= t! \cdot \frac{1}{n!} \cdot (n-t-1)! \\ &= \frac{1}{n} \binom{n-1}{t}^{-1} \end{aligned} \tag{22}$$

The coefficient $w_{ij}^T$ only depends on the cardinality of $T$. Finally, the Shapley value is represented as follows:

$$\phi_i(v) = \Delta_i v(\emptyset) + \sum_{t=0}^{n-2} \frac{1}{n} \binom{n-1}{t}^{-1} \sum_{\substack{j \in N \\ j \neq i}} \sum_{\substack{T \subseteq N \setminus \{i,j\} \\ |T|=t}} I_{ij}(T). $$

$\square$

### C.2 PROOF OF THEOREM 2

**Theorem 2.** *A game $v$ is convex if and only if $I_{ij}(R) \geq 0 \ \forall i,j \in N, \forall R \subseteq N \setminus \{i,j\}$.*

*Proof.* $(\rightarrow)$ **Necessity.**

We show $I_{ij}(R) \geq 0$ for any $i,j \in N$ and $R \subseteq N \setminus \{i,j\}$. Define $S = R \cup \{i\}$ and $T = R \cup \{j\}$. By the convexity of $v$, we obtain

$$\begin{aligned} v(R \cup \{i.j\}) + v(R) &\geq v(R \cup \{i\}) + v(R \cup \{j\}), \\ \Delta_i v(R \cup \{j\}) &\geq \Delta_i v(R). \end{aligned} \tag{23}$$

Thus, it follows that $I_{ij}(R) \geq 0$.

$(\leftarrow)$ **Sufficiency.**

Given $S, T \subseteq N$, set $P = S \setminus T$, $Q = T \setminus S$, and $R = S \cap T$. Let $p, q$ denote the cardinality of $P, Q$, respectively. Choose permutations $\pi \in \Pi(P), \bar{\pi} \in \Pi(Q)$. By Lemma 1, the following equations hold for all $k \in \{1, \cdots, q\}$.

$$\Delta_{\bar{\pi}_k} v(S \cup [\bar{\pi}]_{k-1}) = \Delta_{\bar{\pi}_k} v(R \cup [\bar{\pi}]_{k-1}) + \sum_{i=1}^{p} I_{\bar{\pi}_k, \pi_i}(R \cup [\pi]_{i-1} \cup [\bar{\pi}]_{k-1})$$

$$\geq \Delta_{\bar{\pi}_k} v(R \cup [\bar{\pi}]_{k-1}) \tag{24}$$

Summing over all $k$, we obtain

$$\sum_{k=1}^{q} \left\{ \Delta_{\bar{\pi}_k} v(S \cup [\bar{\pi}]_{k-1}) - \Delta_{\bar{\pi}_k} v(R \cup [\bar{\pi}]_{k-1}) \right\} \geq 0,$$

$$\left\{ v(S \cup [\bar{\pi}]_q) - v(S) \right\} - \left\{ v(R \cup [\bar{\pi}]_q) - v(R) \right\} \geq 0, \tag{25}$$

$$v(S \cup T) - v(S) - v(T) + v(S \cap T) \geq 0.$$

Therefore, the game $v$ is convex.

$\square$

### C.3    PROOF OF THEOREM 3

Let $i \sim u_1(S)$ be the uniform distribution of player $i$ over the set $S$, and $T \sim u_2(t, S)$ be the uniform distribution over subsets $T \subseteq S$ with cardinality $|T| = t$. $\phi(v)$ and $\Delta v(\emptyset)$ denote the Shapley value and the discrete derivatives of all players in vector form, respectively. We define an interaction vector $I_{\cdot j}(T) \in \mathbb{R}^n$ where the $i$-th element is $I_{ij}(T)$ for given $T \subseteq N \setminus \{j\}$. We set $I_{ij}(T) = 0$ when $i = j$ or $i \in T$, ensuring $I_{\cdot j}(T)$ is well-defined for $T \subseteq N \setminus \{j\}$. Then, we obtain the following vectorized form to estimate the Shapley value with interaction vectors.

**Theorem 3.** *The vector of the Shapley value $\phi(v)$ is represented as follows:*

$$\phi(v) = \Delta v(\emptyset) + \sum_{t=0}^{n-2} \mathbb{E}_{j \sim u_1(N), T \sim u_2(t, N \setminus \{j\})}[I_{\cdot j}(T)].$$

*Proof.* Since we set $I_{ij}(T) = 0$ when $i = j$ or $i \in T$, the following equality holds for all $i \in N$:

$$\sum_{\substack{j \in N \\ j \neq i}} \sum_{\substack{T \subseteq N \setminus \{i,j\} \\ |T| = t}} I_{ij}(T) = \sum_{j \in N} \sum_{\substack{T \subseteq N \setminus \{j\} \\ |T| = t}} I_{ij}(T)$$

$$= n \cdot \binom{n-1}{t} \cdot \mathbb{E}_{j \sim u_1(N), T \sim u_2(t, N \setminus \{j\})}[I_{ij}(T)]. \tag{26}$$

By combining Theorem 1 with Equation (26) and vectorizing the result, we obtain the conclusion:

$$\phi(v) = \Delta v(\emptyset) + \sum_{t=0}^{n-2} \mathbb{E}_{j \sim u_1(N), T \sim u_2(t, N \setminus \{j\})}[I_{\cdot j}(T)].$$

$\square$

## D    AN ILLUSTRATIVE EXAMPLE OF UNDERVALUATION

In Figure 4, we provide an example of undervaluation in an image recognition scenario. Pattern scores for image classification are computed within each region, followed by max pooling, and the results are summed to obtain the final output. In this example, the efficiency axiom constrains the total attributions in each region to 7 and 4, respectively. Although Region 1 (the first max pooling) extracts features that significantly contribute to the output, the Shapley value assigns a lower attribution to feature '7' compared to feature '4' in Region 2 (the second max pooling). This occurs due to strong negative interactions within an inefficient coalition, as discussed in Section 4. The proposed API effectively alleviates this undervaluation by eliminating problematic negative interactions, ensuring that feature '7' receives a higher attribution (7) than feature '4' (4).

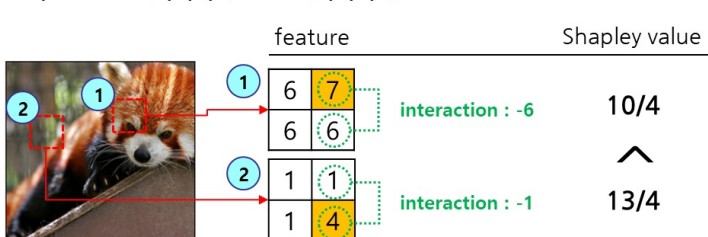

Figure 4: An Example in Image Recognition. Due to negative interactions, feature '7' receives a lower Shapley value than feature '4', which is a counterintuitive result.

# E  ADDITIONAL EXPERIMENTS

## E.1  EXPERIMENT RESULTS FOR COMPLEXITY

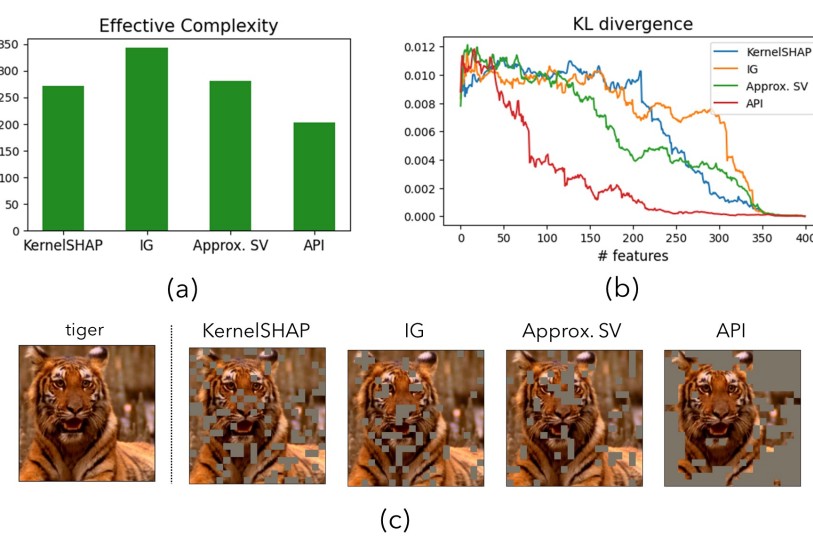

Figure 5: Experiment Results for Complexity. We measured the minimum number of features inserted (in descending order of attribution scores) to achieve the KL divergence between the model output and the original model output below 0.001. API requires fewer features to reproduce the original model output compared to other methods.

We additionally conduct quantitative experiments. We insert the features (i.e., patches) in descending order of attribution scores and compute the output class probability distribution to measure the KL divergence with the original distribution (which includes all features). Then, we identified the minimum number of features required to reduce the KL divergence to below 0.001. Figure 5 (a) shows the average number of features required, with API achieving almost 30% reduction compared to other methods. More specifically, the average numbers are 272.34, 342.94, 281.59, and 203.32 for KernelSHAP, IG, Approximated Shapley value, and API, respectively. This indicates API requires fewer features to reproduce the original model output, effectively capturing the relevant features for the model prediction.

Figure 5 (b) shows the KL divergence with respect to the number of features inserted for a specific image, as depicted in Figure 5 (c). We observe that API leads to a more rapid decrease in KL divergence with relatively fewer features inserted. Specifically, the number of features required to reach the KL divergence of 0.001 is 311, 339, 337, and 203 for KernelSHAP, IG, Approximated Shapley value, and API, respectively. Furthermore, Figure 5 (c) shows a visualization of the features

inserted to achieve the KL divergence of 0.001. With the proposed API, we can reproduce the original decision with fewer irrelevant features inserted compared to other methods.

## E.2 ABLATION STUDY FOR FUNCTION CHOICE ON INTERACTIONS

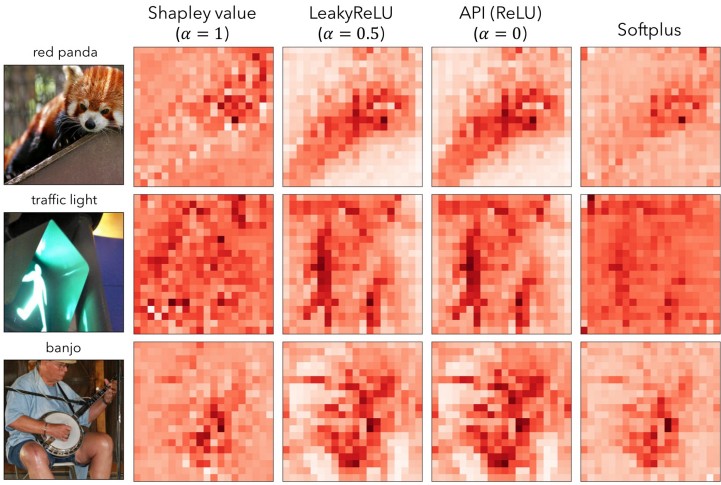

Figure 6: Attribution Results with Different Function Choices on Interactions.

Assigning attributions in the form of interactions in Equation (7) can be seen as decomposing the characteristic function into interaction terms and then transforming these through a function $g$ (e.g. ReLU) to determine the attributions for each player. If $g$ is the identity function, the result is the Shapley value; if it is the ReLU function, the result is API.

The choice of $g$ depends on how the user interprets interactions as attributions in the given task. For instance, if a smoother transition from interaction values to attributions is preferred, a softplus function might be a suitable choice. Figure 6 shows the impact of different function choices in an ImageNet classifier. In this setting, due to the large proportion of negative interactions, even a slight relaxation using the Leaky ReLU function yields results similar to those of the API (albeit with differences in scale). When applying the softplus function, negative interactions are mitigated; however, the differences in positive interactions are also slightly reduced, leading to lower contrast in the final attributions.

## E.3 RUNTIME OF API

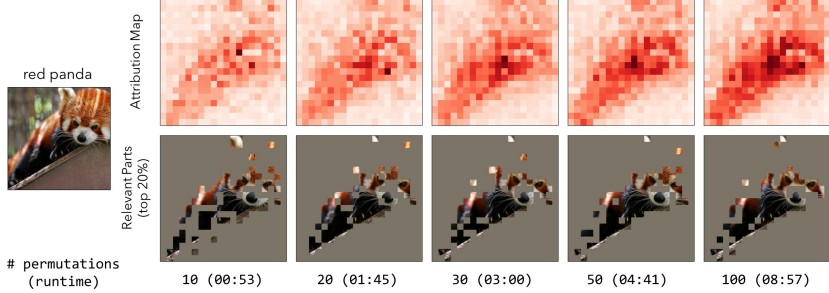

Figure 7: Attribution Results and Runtime with respect to the number of Permutations. API takes around 55 seconds for every 10 permutation samples. Notably, stable results can be achieved with as few as 30 permutations.

Figure 7 shows the runtime of API for different numbers of permutation samples on an ImageNet classifier. We observed that stable results can be achieved with as few as 30 permutations, which

took around 3 minutes on an RTX 6000 GPU. While this is slower compared to IG, it remains comparable to other permutation-based attribution methods grounded in game theory. Since handling interactions inherently involves significant computational costs, accelerating this process through path-based approximations is one of our future research directions.

