# OpenReview forum: "Rethinking Shapley Value for Negative Interactions in Non-convex Games"
_ICLR.cc/2025/Conference — ICLR 2025 Poster_

### Official Review · Reviewer_fJo6 · 2024-10-26

**Soundness:** 2
**Presentation:** 3
**Contribution:** 2
**Rating:** 5
**Confidence:** 4

**Summary:**

The paper reformulates the Shapley value as $\Delta(\emptyset)$  plus a sum of second-order derivatives, which are refered to as interactions in the paper, and argues that ignoring negative second-order derivatives in the summation can provide better feature attribution.

**Strengths:**

Overall, the writing is clear, and the reformulation of the Shapley value seems interesting.

**Weaknesses:**

- Theorem 2 is already a well-known result in the literature, e.g., proposition 2.1 in (Nemhauser et al., 1978).


- In lines 176-177, the authors suggest that a game has a non-empty core if and only if the game is convex.  However, a balanced game is not necessarily convex, and thus this statement is false. A counterexample is $v(\emptyset)=0, v(\\{1\\})=v(\\{2\\})=v(\\{3\\})=3, v(\\{1,2\\})=v(\\{1,3\\})=v(\\{2,3\\})=5$ and $v(\\{1,2,3\\})=10$; the game is not convex as $v(\\{1\\}) -v(\emptyset) > v(\\{1,2\\}) - v(\\{2\\})$, and its core contains $(10/3, 10/3,10/3)$.

- I did not find any *clear* evidence to support the claim that "Our theoretical analysis demonstrates that classical payoff allocation in a cooperative game assumes the convexity of the game" stated in the abstract, which seems to be an overstatment.

- I did not feel that the motivation is compelling enough, please refer to my question.

Minor:
- In lines 133-134, the statement that ''the Shapley value is a solution positioned at the center of the core'' is not strictly correct, because the Shapley value of a game always exists while its core may not if the game is not convex. Besides, I would recommend that the authors include a reference for this statement for those who are not familiar. I know that for a convex game, the Shapley value is the average of $n!$ marginal vectors (though there may be duplicates)  that constitute all the extreme points of the corresponding core, which is convex.

- I feel it may confuse researchers by referring to second-order derivatives as just interactions, because there are already several defined interactions in the literature (Tsai et al. 2023). To fit into the existing studies, I think it would be better to term them as *second-order* interactions.

- The proof of Theorem 1 can be greatly simplified using $\phi_i(v) = \frac{1}{n!}\sum_{\pi} (v(\pi\^{i} \cup \\{i\\}) - v(\pi\^{i}))$ where $\pi^i$ contains all players preceding $i$ in the permutation $\pi$.

- I would recommend that the authors write down the proposed API results in the case study for clear comparison.

- For the case study of sigmoid function, the Shapley values of player 1 and 3 are $2.51$ and $3.73$, respectively.

Nemhauser, G. L., Wolsey, L. A., & Fisher, M. L. (1978). An analysis of approximations for maximizing submodular set functions—I. Mathematical programming, 14, 265-294.

Tsai, C. P., Yeh, C. K., & Ravikumar, P. (2023). Faith-shap: The faithful shapley interaction index. Journal of Machine Learning Research, 24(94), 1-42.

**Questions:**

My major concern is the motivation of filtering out negative second-order derivatives, which the authors refer to as interactions. I do not think that the two simple examples as well as a few experiment results are sufficient. For the Shapley value, it is deemed good because it uniquely satifies the axioms of linearity, null, symmetry and efficiency. For the proposed API, which properties make it desirable?

---

> ### Author Response · Authors · 2024-11-19
> **Rebuttal by Authors**
>
> **Q1. Motivation of filtering out negative interactions (second-order derivatives).**
>
> **A1.** Please refer to the general response provided above for the details.
>
> In summary, the proposed method satisfies the four axioms for convex games, but it no longer satisfies efficiency for non-convex games. If we maintain efficiency in non-convex games, the sum of attributions remains fixed, preventing the compensation for undervaluation from negative interactions—some players will inevitably receive lower attributions. We believe that a reasonable solution need not satisfy efficiency in non-convex games, as the presence of negative interactions implies that the players' coalition is not a fully efficient strategy. To alleviate the undervaluation, our goal is to evaluate each feature/player based on its potential effect to increase the output value by avoiding inefficient scenarios. In this spirit, API was designed to evaluate a player’s causal effect (output changes) from only cooperative behaviors between players.
>
> As explained in General Response, computing API can be interpreted as a two-step process: (1) approximating an original non-convex game with a convex game, and (2) computing the Shapley value on the projected convex game. In choosing a convexified game, our current solution design was guided by two key criteria: (1) positive interactions must be preserved, and (2) the approximated characteristic function should remain as close as possible to the original function. That is why we take max with zero for interaction terms.
>
> In addition, API satisfies two properties
> - The efficiency holds for the revised (convexified) function.
> - Zero attribution guarantees that the corresponding feature cannot improve the output in any case, which does not hold in other attribution methods for non-convex games.
>
> **Response to Comments in Theoretical Parts**
>
> - Related to Theorem 2, we checked the proposition 2.1 in Nemhauser et al.’s work [1] and we will cite it in our manuscript.
> - In the second weakness you pointed out, we acknowledge that we misunderstood the connection between core existence and convexity, and we will revise the corresponding statement accordingly. We assumed that a solution represented by the weighted sum of marginal vectors, including the Shapley value, must always belong to the core since it can be interpreted as the combination of outputs from rational strategies. Initially, we believed that this statement was equivalent to the core existence; however, we later realized this was incorrect. It corresponds to the condition under which the core and the Weber set are equivalent—that is, convexity. We would appreciate it if you could note that this correction does not critically impact our main argument on the issue of negative interactions or our proposed solution to address it. Nonetheless, your feedback has been truly helpful in clarifying and improving our manuscript.
>
> **Response to other comments**
> - Thank you for your valuable feedback. It must be helpful to improve the details of our work. We will incorporate it into the revised manuscript. We will revise some overstated parts and mis-calculated values, and clarify the meaning of interaction which was designed as a second-order derivative in the literature about causal effect [2].
>
> [1] Nemhauser, George L., Laurence A. Wolsey, and Marshall L. Fisher. "An analysis of approximations for maximizing submodular set functions—I." (1978)
>
> [2] Egami, Naoki, and Kosuke Imai. "Causal interaction in factorial experiments: Application to conjoint analysis."  (2019)

---

> ### Comment · Reviewer_fJo6 · 2024-11-25
>
> I have read through the authors' responses, including the global one, and I confirm that the second weakness was not considered in my initial score. I noticed that the authors provide three additional properties for the proposed API, but I could not find any detailed justification for them.
> - If I understand correctly, the authors seem to suggest that the proposed API is equal to calculating the Shapley value of
> \begin{equation}
> v(S)=v(\emptyset)+\sum_{i\in S}\Delta_iv(\emptyset)+\sum_{\substack{i,j\in S, \ i\neq j}} \sum_{T\subseteq S\setminus{i,j}} \frac{1}{s}\binom{s-1}{t}^{-1}\max(I_{ij}(T), 0) .
> \end{equation}
> Although I did not mention it in my initial review, my first impression is that the proposed API seems to first convexify games and then calculate the Shapley value of the convexified games. However, the submission and the responses do not provide any detailed discussion on this.  The above equation seems to refer to this possibility. Could the authors elaborate on this aspect theoretically?
>
> - In addition, in the global response, the authors write "applying ReLU achieves the closest approximation to the original game under these constraints." I find this vague, as "closest" is not rigorously defined. A better choice is to prove that the convexified game $\hat{v}$ is an optimal solution to, e.g., $\min\_{u \in \mathcal{U}} \\|u - v\\|$ where $\mathcal{U}$ contains all the games that satisfy the specified constraints.
>
> - Finally, **there is an issue with the claim provided in the global response** that if the proposed API assigns $0$ to the  $i$-th player, then $\Delta_iv(S)\leq0$ for all $S\subseteq N\setminus {i}$. A counterexample is as follows: let $N = \\{1,2,3\\}$ with $v(\emptyset)=v(\\{1, 3\\})=v(\\{2, 3\\})=0$, $v(\\{2\\})=1$, $v(\\{1\\})=-1$, $v(\\{1,2\\})=3$, $v(\\{3\\})=10$ and $v(\\{1,2,3\\})=-20$. In this case, the proposed API assigns $0$ to player $1$, but $\Delta\_{1}v(\\{2\\}) = v(\\{1,2\\}) - v(\\{2\\}) = 2 > 0$.
>
> **Properties of the proposed API**
>
> Discarding the axiom of efficiency is not a novel idea. For example, the Beta Shapley values proposed by Kwon and Zou (2022a) consider this, and their effectiveness is demonstrated by Kwon and Zou (2022b). That being said, the proposed API is not unique in ignoring the axiom of efficiency. Plus, using the terminology of (Weber, 1988), it is straightforward to verfiy that the proposed API satisfies the axioms of dummy, symmetry and monotonicity. However, all the probabilistic values mentioned by Weber (1998) satisfy these three axioms. Perhaps, there is something else that makes the proposed API unique or desirable?
>
>
>
> Kwon, Y., & Zou, J. (2022a). Beta Shapley: a Unified and Noise-reduced Data Valuation Framework for Machine Learning. In International Conference on Artificial Intelligence and Statistics (pp. 8780-8802). PMLR.
>
> Kwon, Y., & Zou, J. Y. (2022b). Weightedshap: analyzing and improving shapley based feature attributions. Advances in Neural Information Processing Systems, 35, 34363-34376.
>
> Weber, R. J. (1988). Probabilistic values for games. In A. E. Roth (Ed.), The Shapley Value: Essays in Honor of Lloyd S. Shapley (pp. 101–120). chapter, Cambridge: Cambridge University Press.

---

> ### Comment · Reviewer_fJo6 · 2024-11-26
> **The possibility of interpretation using convexification**
>
> Personally, I think that the starting point of this work might be promising and I have been trying to figure out how to make this work more complete in theory. In my view, the proposed method is not well justified theoretically. I already noticed that the proposed API might be interpreted by convexification while reviewing, but it is not a concrete concept because there is no theoretical justification provided in eithier the submission or reponses.
>
> For deriving the convexification, my understanding is that it concerns with the bijective conversion between games and their first- and second-order derivatives.  Precisely, assuming $v(\emptyset)=0$, there are $n + n(n-1) 2\^{n-3}$ variables standing for all first- and second-order derivatives, but the freedom of variables for games is $2\^{n} - 1$. That being said, one conversion is overdetermined and the other is underdetermined. Given this, I am not sure if the interpretation using convexification is possible.
>
> I would be happy to raise my score if the authors can step further in theory for interpreting the proposed API. So far, I did not see anything new in theory compared with the initial submission. This lack of theoretical interpretation makes me feel that the good performance of the proposed API is more of a magic.
>
> **Why convexification is good**
>
> This section is intended purely for discussion. Personally, I think Eq. (1) is not a proper perspective for domenstrating why convexification is good. Despite that, I noticed that the authors have discussed its equivalent definition in the remark right below Eq. (1), which states that $\Delta\_{i}v(S) \leq \Delta\_{i}v(T)$ for every $i \in N$ and $S \subseteq T \subseteq N \backslash \\{ i\\}$. I think this one is more convenient for interpreting why convexification is good. In my own understanding, I think it suggests that it is probably better to fous on the approximate game (best in some sense) where the marginal contribution of one specific feature should not decrease while introduing more other features.

---

> > ### Author Response · Authors · 2024-12-01
> > **Additional Response by Authors**
> >
> > We sincerely appreciate your feedback and suggestions. This communication has provided a valuable opportunity to revisit and further develop our research from both philosophical and theoretical perspectives. Let us begin by addressing the discussion on convexification.
> >
> > **Discussion on Convexification**
> >
> > Our mathematical formulation is inspired by the fact that, just as the Shapley value can be expressed as the average of marginal contribution vectors, it can also be decomposed and represented in terms of interactions. The equation above is derived as the average of $v(S)$ decompositions based on permutations.
> > $$
> > \begin{aligned}
> > v(S)&=\frac{1}{s!}\sum_{\pi\in\Pi(S)}[v(\emptyset)+\sum_{t=1}^s\Delta_{\pi\_t}v([\pi]\_{t-1})] \\\\
> > &=v(\emptyset)+\frac{1}{s!}\sum_{\pi\in\Pi(S)}\sum\_{t=1}^s[\Delta_{\pi\_t}v(\emptyset)+\sum\_{k=1}^{t-1}I\_{\pi\_t\pi\_k}([\pi]\_{k-1})]\\\\
> > &=v(\emptyset)+\sum_{i\in S}\Delta\_iv(\emptyset)+\frac{1}{s!}\sum_{\pi\in\Pi(S)}\sum_{t=1}^s \sum_{k=1}^{t-1}I\_{\pi\_t\pi\_k}([\pi]\_{k-1})
> > \end{aligned}
> > $$
> > When convexifying the original characteristic function, we aimed to preserve positive interactions since our motivation is to avoid undervaluation. Therefore, we considered the following optimal solution (where CG denotes a set of convex games) :
> > $$
> > \begin{aligned}
> > \textsf{argmin}\_{\hat{v}\in\textsf{CG}}&
> > \quad \sum_{S\subseteq N}|v(S)-\hat{v}(S)|
> > \\\\
> > &\textsf{s.t. } \textsf{ReLU}(I_{ij}(T;v))=\textsf{ReLU}(I_{ij}(T;\hat{v})) \quad \forall i,j,T
> > \end{aligned}
> > $$
> > Taking the ReLU on the interaction terms in Equation (5) satisfies the conservation constraints while minimizing changes to the game output. Since a mapping from the game to its interactions is injective, the corresponding characteristic function is unique.
> >
> > However, based on your insightful comment regarding the degree of freedom in selecting interactions, we have realized that additional conditions are required for the existence of the corresponding characteristic function. Specifically, the result of redefining $\hat{v}(S)$ as the sum of consecutive interactions must not depend on the permutation selection (The current characteristic function outputs the average value on permutations). Consequently, to interpret API as a complete convexification, the following condition must hold:
> > $$
> > \begin{equation}
> > \sum_{t=1}^s\sum_{k=1}^{t-1}\textsf{ReLU(}I\_{\pi\_t\pi\_k}([\pi]\_{k-1}))=\sum_{t=1}^s\sum_{k=1}^{t-1}\textsf{ReLU(}I\_{\bar{\pi}\_t\bar{\pi}\_k}([\bar{\pi}]\_{k-1}))\quad \forall\pi,\bar{\pi}\in\Pi(S)
> > \end{equation}
> > $$
> > This observation suggests further study to clarify the necessary & sufficient conditions for reconstructing a game from its first and second derivatives. Currently, we are exploring additional theoretical interpretations that can be presented independently of these conditions, and we hope to identify and share additional insights before the discussion period concludes.
> >
> > **Additional Property of API in Global Response**
> >
> > We have reviewed the counterexample you provided regarding the additional property we proposed and realized that, in designing the statement, we implicitly assumed that $\Delta_i v(\emptyset) \geq 0$. This statement is generally satisfied when the game is 0-normalized, as follows.
> > $$v\_0(S)=v(S)-\sum_{i\in S}v(i)$$
> >
> > **Novelty beyond Efficiency**
> >
> > The studies you suggested provide valuable insights into the constraints of efficiency, particularly regarding the effectiveness of weight distributions based on coalition sizes. However, the focus of our research is on interpreting how interactions between players influence the computation of the Shapley value. Through this investigation, we identified that negative interactions can lead to the issue of attribution undervaluation. Additionally, we found that the existence of negative interactions conceptually conflicts with the origins and principles of the efficiency property, which is why we described it in this manner.

---

> > > ### Comment · Reviewer_fJo6 · 2024-12-02
> > >
> > > Thanks for your response. In terms of convexification, I think it would be better to present concrete proofs instead of speculations. So, I will maintain my score, as I, personally, do not consider it complete in its current form.
> > >
> > > **Additional Property of API**
> > >
> > > I did not see that a zero-normalized game $v\_{0}$ would satisfy $\Delta\_{i}v\_{0}(\emptyset) \geq 0$. A counterexample is: $N=\\{1,2\\}$ with $v\_{0}(\emptyset)=v\_{0}(\\{1\\})=v\_{0}(\\{2\\})=0$ and $v\_{0}(N)=-1$; observe that $v\_{0}(N) - v\_{0}(\\{2\\}) = -1 < 0$.
> > >
> > > **Novelty Beyond Efficiency**
> > >
> > > Although I did not explicitly mention it in my initial review, I have already accounted for the novelty of deriving the Shapley value using first- and second-order derivatives in my initial score. However, I think that there lacks a satisfactory insight for interpreting why setting all the negative second-order derivatives to be $0$ would improve performance in feature attribution.

---

### Official Review · Reviewer_ApX2 · 2024-10-29

**Soundness:** 3
**Presentation:** 2
**Contribution:** 3
**Rating:** 8
**Confidence:** 4

**Summary:**

This paper proposes a modification to the Shapley value for its use in measuring feature attribution of deep learning models. This modification is based on an equivalent reformulation of the Shapley value formula as a weighted sum of pairwise interactions between players (features). When pairwise interactions are non-negative (i.e., a convex game by Theorem 2), Shapley value gives intuitive results and also satisfies desireable conditions from a cooperative game theory perspective, i.e., it lies at the center of the core and satisfies coalitional rationality. However, when interactions are negative, results can be counterintuitive as the authors show through examples; in addition, this coincides with a loss of coalitional rationality. Naturally, the authors suggest clipping interactions to always be non-negative and report the value of this formula instead of the Shapley value. This formula reduces to the Shapley value when the underlying game is convex, but gives more intuitive results in their examples when the game is non-convex. The authors also propose a more tractable estimate using a Taylor approximation and show their approach is competitive with baselines on explainable AI tasks for image and text domains.

**Strengths:**

I think Theorems 1 and 2 that reformulate Shapley value in terms of interactions and relate non-negative interactions to convex games are very insightful. These form the core theoretical discovery that enables the rest of the work. In addition, the case study examples appealing to the reader’s intuition on how negative interactions result in counter-intuitive Shapley value are helpful. Lastly, the Taylor approximation of their approach is competitive with standard baselines and provides intuitive features attributions.

**Weaknesses:**

I see opportunities for improvement both in writing / motivation as well as experiments.

**Writing / Motivation**:
- The 4 axioms that uniquely lead to the Shapley value are mentioned several times but never defined. Any modification to the Shapley value necessarily leads to the loss of one of these axiomatic properties and it should be discussed how this manifests as a limitation of the proposed approach.
- Generally, the paper attempts to argue that Shapley value implicitly assumes a convex game (or least the formation of the grand coalition). I wonder if there is a better argument for the proposed modification. While it is clear that the core makes the assumption of a grand coalition forming, all Shapley value assumes are the 4 axioms, none of which explicitly require any rationality or formation of the grand coalition. Note these axioms are always satisfied regardless of the convexity of the game, and so statements like “Shapley value becomes nothing but taking an expectation formula” on line 189 are hyperbole. In addition, the first remark on line 094 states that the formation of the grand coalition is implicit in cooperative game theory, which is also too strong a statement (for instance, the problem of coalition structure generation obviously does not assume this). It seems the authors are taking the existence of the core as an axiom (core existence <-> convexity <-> non-negative interactions), and that would better serve as the starting point for their argument. Deep learning models typically assume one would make use of all features (i.e., the grand coalition forms), and so it seems sensible to assume this. Continuing this line of thought, could you better argue that the natural characteristic function for DL is one that is convex (i.e., is consistent with the formation of the grand coalition) and your work can be used to implicitly define such an appropriate function via clipping interactions (see questions)?

**Experiments**:
- Given the discussion of the core and Shapley value’s relation to it, why not compare against least-core [1,2] which always exists?
- I would have liked to see a systematic study comparing your exact definition in (6) to your Taylor approximation in (9). How does the approximation degrade? Can you sample a subset of coalitions (features) T and compare the two?
- I would have also liked to see how results in one of the domains (image or text) changes as the number of permutations varies (not just 100 or 300 permutations). How did you select the number of samples?
- Given that computational efficiency was a motivation for the Taylor approximation, can you please report runtimes for the different methods? For instance, IG seemed to perform relatively well yet its unclear if it is generating these results in a runtime that is comparable to your proposed approach.

[1] Benmerzoug, Anes, and Miguel de Benito Delgado. "If You Like Shapley Then You’ll Love the Core." ML Reproducibility Challenge 2022. 2023.

[2] Gemp, Ian, et al. "Approximating the Core via Iterative Coalition Sampling." Proceedings of the 23rd International Conference on Autonomous Agents and Multiagent Systems. 2024.

minor:
- line 098: “forming the grand coalition N is **an** optimal strategy”, e.g., v(S) = 0 is convex yet any coalition is optimal.
- line 112: “from **this** perspective”
- line 175: “we introduce**d** the concept”
- line 266: “It is **the** extension” is too strong. Given Shapley value is derived to be unique from axioms, that is your baseline for strong statements. You would need to prove that API is the only formula satisfying some set of conditions.
- line 359: “we introduce**d** some”
- line 410: “ResNet 50 in Figure ?” missing reference
- Please include more informative captions: for example, lines 413 - 417 can be moved to the caption of Figure 2.

**Questions:**

- Which of the 4 axioms underpinning the Shapley value does your approach no longer satisfy? can you discuss this as a limitation?
- Can applying a ReLU to the interactions also be interpreted as first projecting the game onto the set of convex games?
- Relatedly, can the characteristic function be rebuilt from interaction terms, I(T)? The I(T) matrix appears to be a discrete hessian, so a characteristic function only exists iff I(T) is symmetric for every subset of players T. I(T) appears to be symmetric by definition. Applying ReLU to entries of it should not break symmetry. Therefore, a characteristic function should exist.
- There is some loss of information when applying the ReLU. Do you foresee any limitations? Are there other approaches that might retain more information or have desireable tradeoffs, e.g., a leaky-ReLU?
- line 452: “highlighting the tokens with attribution in the top 20%” - I was expecting to see the same number of tokens highlighted in Figure 3. Line 465 suggests tokens are word level. Why do I not see the same number of words highlighted. If tokens aren’t word level, can you underline the actual tokens?

---

> ### Author Response · Authors · 2024-11-19
> **Rebuttal by Authors**
>
> **Q1. Axioms that our approach no longer satisfies.**
>
> **A1.** Please refer to the general response provided above for the details.
>
> In summary, the proposed method satisfies the four axioms for convex games, but it no longer satisfies efficiency for non-convex games. If we maintain efficiency in non-convex games, the sum of attributions remains fixed, preventing the compensation for undervaluation from negative interactions—some players will inevitably receive lower attributions. We believe that a reasonable solution need not satisfy efficiency in non-convex games, as the presence of negative interactions implies that the players' coalition is not a fully efficient strategy. To alleviate the undervaluation, our goal is to evaluate each feature/player based on its potential effect to increase the output value by avoiding inefficient scenarios. There may be concerns regarding the lack of adherence to the efficiency axiom. However, as discussed in the response to the next question, our method can be interpreted as satisfying efficiency within the context of a convexified characteristic function.
>
> **Q2. Can applying ReLU to the interactions also be interpreted as first projecting the game onto the set of convex games? Can the characteristic function be rebuilt from interaction terms?**
>
> **A2.** Please refer to the general response provided above for the details.
>
> In summary, computing API can be interpreted as a two-step process: (1) approximating an original non-convex game with a convex game, and (2) computing the Shapley value on the projected convex game. Applying ReLU (max with zero) to interaction terms means that attributions are computed by evaluating output changes from only cooperative behaviors between players in the original characteristic function (game), which is equivalent to projecting to convex games. In choosing a convexified game, our current solution design was guided by two key criteria: (1) positive interactions must be preserved, and (2) the approximated characteristic function should remain as close as possible to the original function. Applying ReLU achieves the closest approximation to the original function while preserving positive interactions, leading to its adoption as our design choice. We can explicitly represent the revised characteristic function in interaction terms as follows. API satisfies efficiency in this revised characteristic function.
>
> $$
> v(S)=v(\emptyset)+\sum_{i\in S}\Delta_iv(\emptyset)+\sum_{\substack{i,j\in S, i\neq j}}
> \sum_{T\subseteq S\setminus\{i,j\}}
> \frac{1}{s}\binom{s-1}{t}^{-1}\max(I_{ij}(T),0)
> $$
>
> **Q3. Limitation from loss of information when applying the ReLU.**
>
> **A3.** Removing negative interactions using ReLU is not intended to result in information loss but rather to prevent undervaluation. This approach is appropriate under the assumption that players cooperate to increase the output. For instance, in deep learning classification models, input variables are used to extract evidence that increases the logit probability. However, in scenarios where a decrease in the function's output serves a meaningful result, such as regression problems, eliminating negative interactions could misrepresent the influence of certain variables and they should be carefully handled. Thus, the appropriateness of removing negative interactions depends on the specific context and objectives of the application.
>
> **Q4. The number of highlighted tokens is different.**
>
> **A4.** In Figure 3, we showed only a portion of the original text input since the entire text is too long considering page limit.
>
> **Q5. How results change as the number of permutations varies. Computational efficiency & Runtimes.**
>
> **A5.** We have added an example in the Appendix illustrating the changes in API results and runtime with respect to the number of permutations, using an ImageNet classifier. Empirically, we observed that stable results can be achieved with as few as 30 permutations. On an RTX 6000 GPU, this process took approximately 3 minutes. While this is slower compared to IG, it is comparable to other permutation-based methods grounded in game theory. Handling interactions inherently involves significant computational costs. Accelerating this process through path-based approximations is one of our future research directions.
>
> **Response to other comments**
> - Thank you for your valuable feedback including various suggestions to clarify our work. It must be helpful to improve the details of our work. We will incorporate it into the revised manuscript.

---

> > ### Comment · Reviewer_ApX2 · 2024-11-24
> >
> > Thank you for your responses and thank you for including the additional experiments in A.6. I will increase my score to 7.
> >
> > Regarding my vague comments on "loss of information", I'm more interested, simply out of curiosity, in learning if you can think of a way to intelligently interpolate between Shapley and API. For instance, if you replace ReLU with leaky-ReLU with parameter $\alpha$ controlling the slope for negative values, $\alpha=1$ corresponds to Shapley and $\alpha=0$ corresponds to API. Can you quantify the degradation in efficiency as $\alpha: 1 \rightarrow 0$? More generally, can you think of any reason to replace ReLU with some more exotic, monotonically increasing function, e.g., softplus?

---

> > > ### Author Response · Authors · 2024-11-25
> > > **Rebuttal by Authors**
> > >
> > > Thank you for your positive feedback.
> > >
> > > Additionally, we would like to address your question regarding the impact of $\alpha$ on efficiency when using leaky-ReLU. Let us start by considering an arbitrary characteristic function $v$. It can be expressed as the difference of two convex functions, $v^+$ and $v^-$, as follows:
> > >
> > > \begin{equation}
> > > v=v^+-v^-
> > > \end{equation}
> > >
> > > \begin{equation}
> > > v^+(S)\coloneqq v(\emptyset)+\sum_{i\in S}\Delta_iv(\emptyset)+\sum_{\substack{i,j\in S, i\neq j}}
> > > \sum_{T\subseteq S\setminus\{i,j\}}
> > > \frac{1}{s}\binom{s-1}{t}^{-1}\max (I_{ij}(T),0)
> > > \end{equation}
> > > \begin{equation}
> > > v^-(S)\coloneqq \sum_{\substack{i,j\in S, i\neq j}}
> > > \sum_{T\subseteq S\setminus\{i,j\}}
> > > \frac{1}{s}\binom{s-1}{t}^{-1}
> > > \cdot\{-\min (I_{ij}(T),0)\}
> > > \end{equation}
> > >
> > > This can be derived from the representation of $v$ in terms of interactions, which we provided in the General Response. Specifically, satisfying efficiency means that $\sum_i \phi_i(v) = v^+(N) - v^-(N)$. When $\alpha$ is introduced, this changes to $\sum_i \phi_i(v) = v^+(N) - \alpha \cdot v^-(N)$. Therefore, as $\alpha \in [0, 1]$, the total sum of attributions increases by $(1-\alpha) \cdot v^-(N)$, leading to efficiency degradation. Currently, the API satisfies efficiency with respect to $v^+$, which is the convex approximation of $v$.
> > >
> > > From a different perspective, assigning attributions in the form of interactions can be interpreted as decomposing the characteristic function into interactions and then passing these through an additional function (e.g. ReLU) to allocate attributions to each player. If this function is the identity function, the result is the Shapley value; if it is the ReLU function, the result is the API. The choice of this function may vary depending on the user's interpretation of attribution and the role of interactions in the task.
> > >
> > > For instance, if one desires a softened difference compared to the original interaction values when computing attributions, a softplus function might be chosen. We have added sample results in the Appendix showing the impact of different function choices in an Image classifier. In this task, due to the significant proportion of negative interaction values, even a slight relaxation with a leaky-ReLU yields results similar to those of the API (albeit with differences in scale). When using the softplus function, while the impact of negative interactions is mitigated, the differences in positive interactions also decrease slightly, leading to outcomes with low contrast.
> > >
> > > We hope this clarifies our perspective on your insightful question.

---

> > > > ### Comment · Reviewer_ApX2 · 2024-11-26
> > > >
> > > > Thank you for following-up with additional insight and for running experiments!

---

> > > > > ### Comment · Reviewer_ApX2 · 2024-11-26
> > > > > **Convexification**
> > > > >
> > > > > In case this helps, if you can prove that applying the ReLU results in interaction terms $I$ that actually correspond to a valid characteristic function, I was imagining proving convexification as follows:
> > > > >
> > > > > - Implicitly define a distance measure between $v$ and $\hat{v}$ such that $D(v, \hat{v}) = \sum_T \sum_{ij} \vert I_{ij}(T) - \hat{I}_{ij}(T) \vert$ where $\hat{I}$ denotes the interaction terms associated with $\hat{v}$.
> > > > > - The projection of $v$ onto the set of convex games is then $v^* = \arg\min\_{v’ \in CV} D(v, v’)$ where $CV$ is the set of convex characteristic functions.
> > > > > - By inspection, clipping all negative interactions achieves the global min (assuming you prove as I said above that $v^*$ is a valid characteristic function).

---

> > > > > > ### Author Response · Authors · 2024-12-01
> > > > > > **Additional Response by Authors**
> > > > > >
> > > > > > Thank you for providing additional comments to help advance our research. Your feedback, along with that of reviewer fJo6, has offered us a valuable opportunity to revisit and develop our work from both philosophical and theoretical perspectives.
> > > > > >
> > > > > > Our method can be interpreted as identifying a characteristic function that preserves contributions from positive interactions while minimizing changes to the game output, as described below.
> > > > > > $$
> > > > > > \begin{aligned}
> > > > > > \textsf{argmin}\_{\hat{v}\in\textsf{CG}}&
> > > > > > \quad \sum_{S\subseteq N}|v(S)-\hat{v}(S)|
> > > > > > \\\\&\textsf{s.t. } \textsf{ReLU}(I\_{ij}(T;v))=\textsf{ReLU}(I\_{ij}(T;\hat{v})) \quad \forall i,j,T
> > > > > > \end{aligned}
> > > > > > $$
> > > > > > However, we have realized that additional conditions are required for the existence of the valid characteristic function. This is because the degree of freedom is higher in the space of interactions than that of (valid) games. When decomposing $v(S)$ into interactions for designing the characteristic function, we implicitly assumed that the sum of consecutive (revised) interactions remains consistent regardless of the permutation, which is necessary for the validity of a game. The current characteristic function outputs the average value on permutations. Consequently, to interpret API as a complete convexification, the following condition must hold:
> > > > > > $$
> > > > > > \sum_{t=1}^s\sum_{k=1}^{t-1}\textsf{ReLU(}I\_{\pi\_t\pi\_k}([\pi]\_{k-1}))=\sum_{t=1}^s\sum_{k=1}^{t-1}\textsf{ReLU(}I\_{\bar{\pi}\_t\bar{\pi}\_k}([\bar{\pi}]\_{k-1}))\quad \forall\pi,\bar{\pi}\in\Pi(S)
> > > > > > $$
> > > > > > This observation suggests further study to clarify the necessary & sufficient conditions for reconstructing a game from its first and second derivatives. Currently, we are exploring additional theoretical interpretations that can be presented independently of these conditions, and we hope to identify and share additional insights before the discussion period concludes.

---

### Official Review · Reviewer_ypS5 · 2024-11-04

**Soundness:** 3
**Presentation:** 3
**Contribution:** 2
**Rating:** 6
**Confidence:** 3

**Summary:**

The paper identifies negative interactions to be an issue for methods that quantify feature attribution by approximating the Shapley value. The paper establishes the equivalency between the convexity of the game and the absence of negative interactions between features, and that the Shapley value can only meaningfully represent the causal effect of a feature’s contribution for convex games. The paper proposes Aggregated Positive Interactions, an extension of the Shapley value to non-convex games that only accounts for positive interactions between features. The papers proposes to estimate Aggregated Positive Interactions using sampling interactions, and show empirical evidence on a few examples in image classification (VGG19 and ResNet50 trained on ImageNet) and sentence classification (BERT trained on the IMDB Review dataset).

**Strengths:**

- The paper is well-written. The problem with using Shapley value for deep learning is rigorously discussed, and the proposed approach is well-motivated.
- A proposed method is simple and intuitive. An efficient estimator (using gradient) is presented to lower the computation cost.

**Weaknesses:**

- The experimental evidence for the effectiveness the proposed Aggregated Positive Interactions is not strong enough. For the image classifiers, only results on a few image examples are shown for two models.. In the language model experiment, only one positive and negative sentence examples are shown for one model. Although the examples are quite illustrative, it’s unclear if they are cherry-picked.
- It’s unclear how scalable the approach is. See questions about its computation cost and sample efficiency.

**Questions:**

- In Algorithm 1, it’s not exactly uniform sampling all permutations $\pi$, as each sample is reused for all $t$. How much of a computation burden is it to resample $\pi$ in the inner loop? And how much effect does it have in the performance?
- Could you discuss the computation cost & sample efficiency for the proposed approach? Since you are using gradient information to approximate the interaction, I would expect the estimator to be quite scalable. It’s odd that you have to convert the images to 20x20 patches for feasible computation.

Minor comments:

- Line 296: I think it’s inaccurate to call (7) an unbiased estimator of the Shapley value. The expectation represents the exact Shapley value. You might want to explicitly provide the Monte Carlo estimator in equation.
- It will be helpful to briefly define what “feature” refers to in this paper. It seems to refer to components of the data in this paper, but in other literature feature often refers to learned representations.
- Line 410: the reference to Figure 2 is not displayed properly.

---

> ### Author Response · Authors · 2024-11-19
> **Rebuttal by Authors**
>
> **Q1. It’s not exactly uniform sampling all permutations $\pi$. How much of a computation burden is it to resample $\pi$ in the inner loop?**
>
> **A1.** In Algorithm 1, the probability of constructing $[\pi]\_{t+1}, [\pi]\_{t}$ by resampling a new permutation $\pi$ at each time step $t$ is equivalent to sampling a permutation beforehand and using $[\pi]\_{t+1}, [\pi]\_{t}$ at each $t$. The latter approach is more efficient as it eliminates the need to resample a permutation for every $t$. Additionally, backpropagation is implemented once at each $t$, whereas the former approach requires it twice. This technique is similarly employed in Castro et al.'s work [1] for efficiently approximating the Shapley value using permutation sampling.
>
> [1] Castro, Javier, Daniel Gómez, and Juan Tejada. "Polynomial calculation of the Shapley value based on sampling." (2009)
>
> **Q2. Computational cost & sample efficiency for the proposed method.**
>
> **A2.** Our algorithm requires $O(n)$ backpropagations per permutation to compute the gradients. This is a significant improvement over directly calculating interactions, which would require $O(n^2)$ forward passes, making such computations prohibitively expensive. The proposed approximation method addresses this computational challenge.
>
> However, permutation-based methods inherently involve high computational costs despite providing robust game-theoretic justification. This is because inserting players (or elements) sequentially necessitates $N$ evaluations. In our study, we achieve computational efficiency comparable to conventional permutation-based methodologies, even while incorporating interactions. For the individual pixel-level application, a path-based approach would likely be suggested, which we consider as our future work.
>
> **Q3. Few experimental results.**
>
> **A3.** To demonstrate the generality of our results, we conducted additional quantitative experiments inspired by the effective complexity metric proposed by Nguyen et al. [2]. Specifically, we evaluated the change in output probability for an ImageNet classifier as features (patches) were added in descending order of attribution scores. If the attributions accurately identify information critical to the model's decision-making, a small number of features should suffice to reproduce the output probability observed with all features included.
>
> In this context, we measured the number of features required for the KL divergence between the original and reconstructed probabilities to drop below 0.001. The average complexity is summarized as follows. Our method achieves almost 30% reduction in the number of features required to reproduce the original decision output compared to alternative methods. We will include this evaluation in the Appendix of the revised paper.
>
> - KernelSHAP : 272.34
> - Integrated Gradients : 342.94
> - Approximated Shapley value : 281.59
> - Aggergated Positive Interactions : 203.32
>
> [2] Nguyen, An-phi, and María Rodríguez Martínez. "On quantitative aspects of model interpretability." (2020).
>
> **Response to other comments**
> - Thank you for your valuable feedback. It must be helpful to improve the details of our work.  We will incorporate it into the revised manuscript.

---

> > ### Comment · Reviewer_ypS5 · 2024-11-26
> >
> > Thank you for your responses! My concerns about the complexity is well addressed, and I appreciate the additional experiments. I've raised the score.

---

### Official Review · Reviewer_kE5C · 2024-11-05

**Soundness:** 3
**Presentation:** 4
**Contribution:** 3
**Rating:** 5
**Confidence:** 2

**Summary:**

This paper extended Shapley Value from convex games into non-convex games. The paper argues that traditional Shapley Value is only a fair and reasonable allocation when then underlying game is balanced or in other words, convex. Therefore, it is not a well-defined concept when non-convexity is involved. To address this issue, the paper proposed a variation of traditional Shapley Value called Aggregated Positive Interaction (API) that decomposes the total utility into pairwise interactions between players. The authors then provided an approach to estimate API using permutation sampling, which proves to be an unbiased estimator. Furthermore, the authors provided approximated method to compute API more efficiently. Experiments demonstrated the effectiveness of the proposed approach.

**Strengths:**

The paper extended traditional Shapley Value to non-convex games, which provided a way for the community to attribute contributions of features in non-convex settings. Given that deep learning models are widely used so far, the method in this paper would of be great importance to general AI and ML community.

The paper provided theoretical justification of the proposed API attribution method, including proof of relationship between traditional Shapley Value and convexity, unbiasedness of API estimation etc. The theoretical contribution of this paper is novel.

The presentation of this paper is nice - easy to read and understand. The case study is interesting and really helps readers understand why traditional Shapley Value has certain limitations.

**Weaknesses:**

The major weakness of the paper is that I don't see a clear and strong justification for why the API is designed this way, specifically why we need to take max with 0. Does the number 0 really matter? Can we take other numbers here? Traditional Shapley Value is supported by those 4 axioms, and they serve as a strong justification for the design of Shapley Value. However, for API, the utility of each individual may not sum up to the total utility. In this case, I am wondering why API has to be designed as in this paper? Does other design work like taking max with another number?

**Questions:**

I am wondering why API has to be designed as in this paper? Does other design work like taking max with another number instead of 0?

---

> ### Author Response · Authors · 2024-11-19
> **Rebuttal by Authors**
>
> **Q1. The utility of each individual may not sum up to the total utility. Why is API designed as in this paper? Does other design work like taking max with another number instead of zero?**
>
> **A1.** Please refer to the general response provided above for the details.
>
> In summary, if we maintain efficiency in non-convex games, the sum of attributions remains fixed, preventing the compensation for undervaluation from negative interactions—some players will inevitably receive lower attributions. To alleviate the undervaluation, our goal is to evaluate each feature/player based on its potential effect to increase the output value by avoiding inefficient scenarios. In this spirit, API was designed to evaluate a player’s causal effect (output changes) from only cooperative behaviors between players.
>
> As explained in General Response, computing API can be interpreted as a two-step process: (1) approximating an original non-convex game with a convex game, and (2) computing the Shapley value on the projected convex game. In choosing a convexified game, our current solution design was guided by two key criteria: (1) positive interactions must be preserved, and (2) the approximated characteristic function should remain as close as possible to the original function. That is why we take max with zero for interaction terms.
>
> In addition, API satisfies two properties
>
> - The efficiency holds for the revised (convexified) function.
> - Zero attribution guarantees that the corresponding feature cannot improve the output in any case, which does not hold in other attribution methods for non-convex games.

---

### Meta-Review · Area_Chair_cXnb · 2024-12-21

**Metareview:**

The paper proposes a modification to the Shapley value for its use in measuring feature attribution of deep learning models. This modification is based on an equivalent reformulation of the Shapley value formula as a weighted sum of pairwise interactions between players (features). The authors suggest clipping interactions to always be non-negative and report the value of this formula instead of the Shapley value. This approach gives same results for convex settings (as the standard Shapley value) but also gives more intuitive results even when the game is non-convex (the authors provide extensive examples). The reviewer with the highest expertise championed the paper and the other reviewers were either slightly positive or slightly negative (i.e., the rest of the reviewers believed that this is a borderline paper). We recommend weak acceptance and suggest the authors to address all the comments of reviewer ApX2 in the camera ready version.

**Additional Comments On Reviewer Discussion:**

The reviewer with the most detailed review and highest expertise was quite positive about the paper and the other reviewers believe that it is a borderline paper.

---

### Decision · Program_Chairs · 2025-01-22

Accept (Poster)